# Automatic Detection of Inland Water Bodies along Altimetry Tracks for Estimating Surface Water Storage Variations in the Congo Basin

Frédéric Frappart [1,2,*], Pierre Zeiger [1], Julie Betbeder [3], Valéry Gond [3], Régis Bellot [3,4], Nicolas Baghdadi [5], Fabien Blarel [1], José Darrozes [6], Luc Bourrel [6] and Frédérique Seyler [7]

1    LEGOS, Université de Toulouse, CNES, CNRS, IRD, UPS—14 Avenue Edouard Belin, 31400 Toulouse, France; pierre.zeiger@legos.obs-mip.fr (P.Z.); fabien.blarel@legos.obs-mip.fr (F.B.)
2    INRAE, UMR1391 ISPA, 33140 Villenave d'Ornon, France
3    CIRAD, Forêts et Sociétés, 34398 Montpellier, France; julie.betbeder@cirad.fr (J.B.); valery.gond@cirad.fr (V.G.); regis.bellot@orange.fr (R.B.)
4    IGN, 94160 Saint-Mandé, France
5    INRAE, UMR TETIS, University of Montpellier, 500 rue François Breton, CEDEX 5, 34093 Montpellier, France; nicolas.baghdadi@teledetection.fr
6    GET, Université de Toulouse, CNRS, IRD, UPS—14 Avenue Edouard Belin, 31400 Toulouse, France; jose.darrozest@get.omp.eu (J.D.); luc.bourrel@ird.fr (L.B.)
7    ESPACE-DEV, Université Montpellier, IRD, Université Guyane, Université Réunion, Université Antilles, Université Avignon, 500 rue Jean-François Breton, 34393 Montpellier, France; frederique.seylert@ird.fr
*    Correspondence: frederic.frappart@legos.obs-mip.fr

**Abstract:** Surface water storage in floodplains and wetlands is poorly known from regional to global scales, in spite of its importance in the hydrological and the carbon balances, as the wet areas are an important water compartment which delays water transfer, modifies the sediment transport through sedimentation and erosion processes, and are a source for greenhouse gases. Remote sensing is a powerful tool for monitoring temporal variations in both the extent, level, and volume, of water using the synergy between satellite images and radar altimetry. Estimating water levels over flooded area using radar altimetry observation is difficult. In this study, an unsupervised classification approach is applied on the radar altimetry backscattering coefficients to discriminate between flooded and non-flooded areas in the Cuvette Centrale of Congo. Good detection of water (open water, permanent and seasonal inundation) is above 0.9 using radar altimetry backscattering from ENVISAT and Jason-2. Based on these results, the time series of water levels were automatically produced. They exhibit temporal variations in good agreement with the hydrological regime of the Cuvette Centrale. Comparisons against a manually generated time series of water levels from the same missions at the same locations show a very good agreement between the two processes (i.e., RMSE $\leq$ 0.25 m in more than 80%/90% of the cases and R $\geq$ 0.95 in more than 95%/75% of the cases for ENVISAT and Jason-2, respectively). The use of the time series of water levels over rivers and wetlands improves the spatial pattern of the annual amplitude of water storage in the Cuvette Centrale. It also leads to a decrease by a factor of four for the surface water estimates in this area, compared with a case where only time series over rivers are considered.

**Keywords:** radar altimetry; wetlands; surface water storage; Congo

## 1. Introduction

Floodplains and wetlands cover at least 12.1 $\times$ 10$^6$ km$^2$ (~8%) of the land surfaces of the Earth [1,2]. They play a major role in the water cycle through river flow variability, flood mitigation, groundwater recharge and water quality improvement [2–4]. They were identified as one of the most productive ecosystems as well as a major contributor of biodiversity within a landscape [5–10]. They also have an important role in the global

carbon cycle as 16 to 33% of the soil carbon pool is stored in the wetlands [11,12] and as 20 to 25% of the methane emissions originated from the wetlands [13–16].

In spite of their importance, the water stored in floodplains and wetlands and its temporal variations are poorly known from regional to global scales. Before the launch of NASA/CNES Surface Water and Ocean Topography (SWOT) in 2022, which will provide surface water elevation over inland water bodies at a spatial resolution of 100 m [17], anomalies of surface water storage are currently derived by: (i) estimating water level changes using an interferometric synthetic aperture radar (SAR) (InSAR) [18,19], (ii) filling a digital elevation model (DEM) with surface water extent estimates from remotely sensed observations through a hypsometric curve [20,21], (iii) combining surface extent products derived from satellite images with radar altimetry based water levels [22,23], or (iv) solving the water balance equation combining various remotely sensed observations (i.e., anomalies of terrestrial water storage (TWS) from the Gravity Recovery And Climate Experiment (GRACE), water levels from radar altimetry, rainfall from Global Precipitation Climatology Project (GPCP), shuttle radar terrain model (RTM), digital elevation model (DEM), synthetic aperture radar (SAR) from Japanese Earth Resources -1 (JERS-1) and multi-spectral from MODerate resolution imaging spectroradiometer (MODIS) images) [24,25]. The latter approach needs the creation of a network of altimetry-based stations of water levels or virtual stations (VSs). Currently, these VSs are mostly obtained through a manual process using dedicated softwares such as the Virtual ALtimetric Stations (VALS) software [26], the Multi-mission Altimetry Processing Software (MAPS) [27,28], or the Altimetry Time Series (AlTiS) software [29], except for large rivers and lakes. This processing is time consuming and does not allow a complete coverage of extensive areas such as the wetlands present in the large river basins, especially when considering a large number of satellite missions. Automatic processing to produce RA-based water levels relies on the identification of the cross-section between a permanent waterbody and the RA ground-tracks (e.g., [30,31]). Therefore, these processes are applied to cross-sections located on rivers, lakes and reservoirs.

Yet, several studies already shown that radar altimetry (RA) can be used for the monitoring of land surface properties. Spatio-temporal changes in radar altimetry backscattering coefficients ($\sigma_0$) were related to land cover types (ice, arid and semi-arid areas, wetlands, forests, ... ) and hydrological changes (floods, soil moisture, ... ) from regional to global scales [32–37]. Over West Africa, signature of floods under the dense vegetation canopy of the Congo River Basin was observed at S, C, Ku and Ka bands through an increase in the backscattering coefficient of various altimetry missions (i.e., ENVISAT, Jason-2 and SARAL) [36]. These results are very encouraging for using the backscattering coefficient to automatically identify radar altimetry measurements over water under dense vegetation cover.

The objectives of this study are to (i) demonstrate the possibility to identify water under the altimeter ground-tracks using the backscattering coefficient, important but poorly used information derived from the RA echo [38], (ii) automatically create time series of water levels over inland waterbodies, including floodplains and wetlands which are not monitored, (iii) to measure the impact of the densification of the networks of altimetry-based water levels on the surface water estimates currently performed combining nadir RA and satellite images. This assessment was achieved on the Cuvette Centrale of Congo, applying an unsupervised classification technique (k-means) to radar altimetry backscattering coefficients acquired at Ku-band (ENVISAT and Jason-2) to automatically discriminate acquisitions made over water bodies (e.g., open water such as rivers, ponds, oxbow lakes, and lakes and water under vegetation as forested floodplains and wetlands) in the extensive floodplain area of the Cuvette Centrale in the Congo River basin. The optimal selection of the cluster numbers was performed using the Calinski–Harabasz criterion [39]. The results of the clustering technique were validated against an unsupervised classification of the Cuvette Centrale validated using several data sources [40]. Based on the results of the clustering technique, the time series of water levels were generated at the cross-sections

with the river network and over the floodplains to construct a network of VSs. Over rivers, the resulting time series of water level were compared, due to the lack of in situ data in this area, this study used the altimetry-based water stages from the Hydroweb database [41]. Then, water level maps were obtained combining surface water extent from PALSAR-1 images (see [40] for details about the processing of the SAR images) and altimetry-based water levels over rivers and floodplains.

## 2. Study Area

The Congo River Basin (Figure 1a) is the largest drainage basin in Africa. Its shape is almost circular, covering an area of ~4 million km$^2$ [42]. It is the second river in the world, after the Amazon River, in terms of discharge (~40,000 m$^3 \cdot$s$^{-1}$) [43]. Its main waterway is the Congo River, with a length of ~6650 km from its source in the southeast of the Democratic Republic of Congo to the Atlantic Ocean [44]. The Congo River Basin is mainly located in the equatorial and tropical savanna climate area according to the Köppen–Geiger climate classification [45], but also in the humid subtropical climate in its southern part, and, marginally, in the semi-arid and desert climates in the east (Figure 1a).

The Cuvette Centrale is a vast floodplain located in the center of the Congo River Basin which extends from 3° S to 3° N in latitude and from 16° E to 22° E in longitude (Figure 1b). It covers an area of 1,176,000 km$^2$ with a wetland extent of 360,000 km$^2$ (32% of its area) [46]. It is the remnant of a lake which occupied the area during the Tertiary geological period that is now surrounded by mountains and plateaus [47]. As this area is difficult to access, the vegetation has been poorly studied [48]. It is mostly covered with tropical evergreen forests, and vegetation adapted to soils saturated with water (e.g., flooded forests and inundated grasslands) [40,46,49,50]. In the Cuvette Centrale are located the confluences between the Congo River and two of its major northern tributaries, the Ubangi River, and downstream, the Sangha River. The annual rainfall in the Cuvette Centrale ranges from 1400 to 1800 mm·year$^{-1}$ and the potential evapotranspiration reaches 1280 mm·year$^{-1}$ [51]. The topography is flat with an average slope lower than 7 cm·km$^{-1}$ between Kisangani and Kinshasa [42], on the 50 km upstream is the outlet of the Kasai River [52] and on the 450 km upstream, is the downstream part of the Ubangi River [53]. The Cuvette Centrale is covered with sandy lacustrine Quaternary sediments where are located the forested wetlands [54,55]. These forested wetlands are subject to the flood pulse of the Congo River [56]. Water levels of the Congo River are characterized by a bimodal flooding pattern, with a main peak of high water in November–December, and a secondary one in April–May, a more pronounced low water stage in August and one of lower intensity occurring in February–March in the Cuvette Centrale. The hydrological regime of the Ubangi (main tributary in the north of the Cuvette Centrale) is characterized by a high water period peaking in November and low water levels in March [57]. Permanently inundated areas are located alongside the rivers. Other areas are inundated either during both flood events (discontinuously during a long time period of 6–7 months) or only during the maximum of the largest inundation event occurring in November–December (during a shorter time period of 2–3 months). More details about the Cuvette Centrale, its vegetation and hydrological regime can be found in [40].

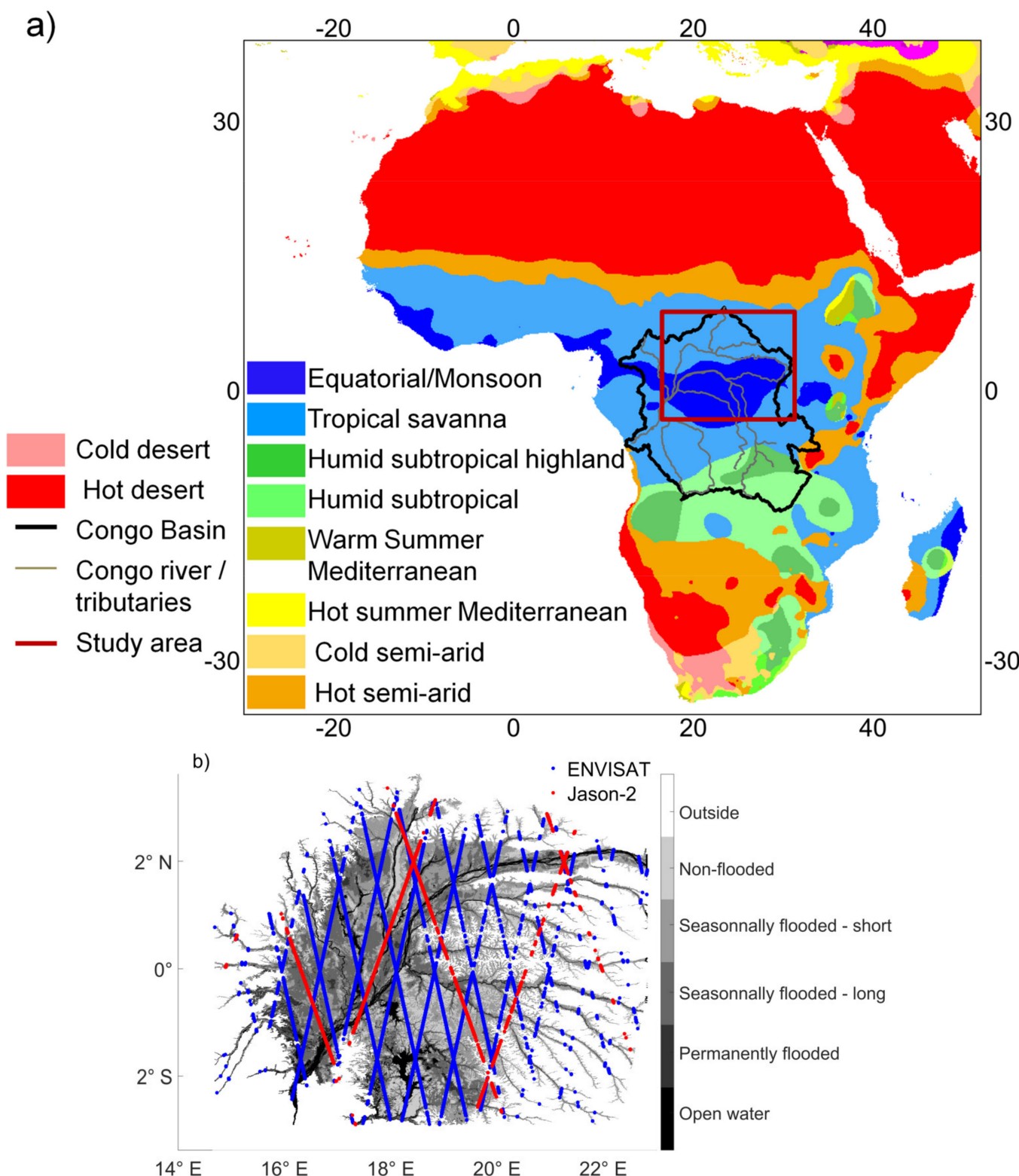

**Figure 1.** (**a**) The Congo River Basin and its location in the climate zones of Africa from the updated Köppen–Geiger classification [43], the red rectangle encompasses the Cuvette Centrale; (**b**) locations of the radar altimetry ground-tracks (Envisat in red, Jason-2 in blue) inside the Cuvette Centrale. Land types come from [38].

## 3. Datasets

### 3.1. Radar Altimetry Data

Data acquired by two altimetry missions (ENVISAT and Jason-2) were considered in this study. These two missions were operating on their nominal orbits from March 2002 to October 2010 and from June 2008 to September 2016 respectively. These two missions were chosen because they were operating when PALSAR-1 onboard ALOS-1 was in operation (see Section 3.2). A complete description of these two altimetry missions can be found in [58] and in [59]. In this study, the following parameters, available at high frequency (i.e., at 18 Hz for ENVISAT and 20 Hz for Jason-2, that is to say ~350 m along the altimetry ground-tracks), were used: the time and geographical locations of acquisition, the distance between the satellite and the surface or range, the orbit of the satellite, the corrections applied to the range to account for the delays caused by the path through the atmosphere (ionosphere, dry and wet atmosphere corrections) and some geophysical effects (solid earth and pole tides), the geoid model and the backscattering coefficient ($\sigma_0$) at Ku-band. Radar altimeter ranges and backscattering coefficients at Ku-band are derived from the offset center of gravity (OCOG) retracking algorithm [60], found to be well-suited for land surface studies in several previous studies (e.g., [26,61]). All these parameters, contained in the geophysical data records (GDR) of each mission, are made available by Centre de Topographie des Océans et de l'Hydrosphère (CTOH) [62], as well as indexes of the normalized tracks [63,64] that were used to compute statistics along the tracks (see Section 4.2). The location of the ground-tracks in the Cuvette Centrale is presented in Figure 1b.

### 3.2. Land Cover Map of the Cuvette Centrale

It results from the merging of two unsupervised classifications obtained using the k-means clustering technique:

(i)     From the enhanced vegetation index (EVI) [65] from the L3 Global 500 m 16-Day moderate resolution imaging spectroradiometer (MODIS) onboard the NASA Terra satellite product (MOD13A1) from 2001 to 2009;

(ii)    From a series of six PALSAR-1 images at 100 m of spatial resolution acquired at L-band, in HH polarization and at various viewing angles (from 18° to 43°), in ScanSAR mode by the phased array type synthetic aperture radar (PALSAR) sensors onboard the Advanced Land Observing Satellite (ALOS) on 7 September 2007, 25 October 2008, 25 January 2009, 27 April 2009, 13 December 2009, and 15 March 2010.

Four classes of forest types and four classes of hydrological status were obtained from the Thorndike index [66]. Through cross-comparisons with several external datasets, the topographic map of northern Congo, ICESat lidar data of elevation and canopy height and aerial photographs, five classes were identified attributed to:

(i)     Open water;

(ii)    Permanently flooded forests;

(iii)   Seasonally flooded forests during the two Congo River flood pulses and located alongside the river;

(iv)    Seasonally flooded forests during a short time corresponding to the maximum of the largest flood pulse, farther from the river;

(v)     Non-flooded forests.

More details about the classification results can be found in [40].

### 3.3. Altimetry-Based Time Series of Water Levels

As no in situ data of water stage are available in the study area during the altimetry period, time series of water levels derived from ENVISAT and Jason-2 data over rivers in the Cuvette Centrale were used to validate the automatic estimates of river heights derived from the unsupervised classification applied to the radar altimetry backscattering coefficients. They were obtained from the Hydroweb database [41]. In this study, 24 and

32 time series from ENVISAT (35-day repeat period) and Jason-2 (10-day repeat period), respectively, were used. The length of the cross-sections between the river and the altimetry tracks range from 100 to 150 m in the upstream parts of the Cuvette Centrale to 10 km in its downstream part.

## 4. Methods

### 4.1. Radar Altimetry Data Pre-Processing

Radar altimetry data were preprocessed in order to:

(i)   Derive the altimeter height (h) from the parameters contained in the GDR using the following equation (e.g., [67]):

$$h = H - R - \Sigma(\Delta R_{propagation} + \Delta R_{geophysical}) + N, \tag{1}$$

where H is the height of the center of mass of the satellite above the ellipsoid estimated using the Precise Orbit Determination (POD) technique, R is the altimeter range (i.e., the distance from the center of mass of the satellite to the surface taking into account instrumental corrections equals to $c\Delta t/2$ where c is the velocity of light in the vacuum and $\Delta t$ is the two-way travel time of the electromagnetic wave emitted by the radar), $\Delta R_{propagation}$ and $\Delta R_{geophysical}$ are the sum of the geophysical and environmental corrections applied to the range, respectively, and N is the geoid.

The corrections to apply to the range for the propagation are the following:

$$\Delta R_{propagation} = \Delta R_{ion} + \Delta R_{dry} + \Delta R_{wet}, \tag{2}$$

where $\Delta R_{ion}$ is the atmospheric refraction range delay due to the free electron content associated with the dielectric properties of the ionosphere, $\Delta R_{dry}$ is the atmospheric refraction range delay due to the dry gas component of the troposphere, $\Delta R_{wet}$ is the atmospheric refraction range delay due to the water vapor and the cloud liquid water content of the troposphere.

And for the geophysical effects:

$$\Delta R_{geophysical} = \Delta R_{solid \cdot Earth} + \Delta R_{pole}, \tag{3}$$

where $\Delta R_{solid \cdot Earth}$ and $\Delta R_{pole}$ are the corrections accounting for crustal vertical motions due to the solid Earth and pole tides, respectively.

(ii)   Obtain along-track time series of backscattering coefficient presenting no missing data in entry of the clustering method. To do so, monthly climatologies (i.e., the average of all data from each month averaged over the whole study period) of backscattering at Ku-band for every normalized index along the altimetry ground-tracks were computed.

### 4.2. Radar Altimetry Data Clustering

K-means clustering is a statistical technique designed to assign objects to a fixed number of groups or clusters based on the analysis of a set of specified variables so that the within-cluster sum of squares is minimum [68]. The initial cluster centers K are selected and then iteratively refined:

(i)   Each object is assigned to its closest cluster in terms of distance to the center of the cluster;
(ii)   At every iteration, each cluster center or centroid is updated to be the average of the member of the cluster.

This process is repeated until convergence.

K-means++ algorithm, which improves the quality of the final solution in terms of intra-cluster distances [69] was used. It benefits from repeated new initial cluster centroid positions. The number of times to repeat clustering using the new initial cluster centroid

positions was set to 5 and the maximum number of iterations before convergence to 100. The distance used for the k-means approach is the Euclidean distance (Ed).

In this study, the k-means approach is used to perform an unsupervised classification of the radar altimetry data using the seasonality of the backscattering at Ku-band. Data used as an input of the k-means clustering are climatological monthly variations in the backscattering at Ku-band for every normalized index along the altimetry ground-tracks [37]. The average ($^{-}$) and standard deviation (*std*) of the radar altimetry backscattering coefficients ($\sigma_0$) are computed on every along-track grid-point as follows [36,37]:

$$\overline{\sigma_0}\,(dB) = 10\log_{10}\left(\overline{10^{\,\sigma_0/10}}\right), \tag{4}$$

$$std(\sigma_0)\,(dB) = 10\log_{10}\left(1 + \frac{std\left(10^{\,\sigma_0/10}\right)}{\overline{10^{\,\sigma_0/10}}}\right), \tag{5}$$

The optimal number of clusters was determined using the Calinski–Harabasz criterion ($I_{CH}$) and the silhouette coefficient (6 and 7, respectively). $I_{CH}$ or variance ratio criterion was defined as the between-cluster variance and the overall within-cluster variance [39]:

$$I_{CH} = \frac{N_S - C}{C - 1}\frac{\sum_{i=1}^{C} d(u_i, U)}{\sum_{i=1}^{C}\sum_{x_j \in CL_i} Ed\left(x_j, u_i\right)}, \tag{6}$$

where $N_S$ is the number of samples, $C$ the number of clusters, $Ed$ the Euclidian distance between two elements, $u_i$ the $i$[th] centroid, $U$ the center of gravity of the whole dataset, $CL_i$ the $i$[th] cluster, $x_j$ is the $j$[th] element of the dataset. $I_{CH}$ is the maximum value obtained for a given maximum number of clusters $C$. Time series of water levels were then obtained computing the median of the consecutive acquisitions from river and wetland classes. In the case of long continuous classes, time series were computed on a maximum distance of 5 km.

The silhouette coefficient provides an assessment of partitioning validity and can be used for determining the optimal number of classes [70].

The validity of the clustering is measured by computing the similarity or cohesion of each cluster member and its separation to the other clusters. The silhouette coefficient for the $i$[th] cluster member is defined as follows:

$$s(i) = \frac{b(i) - a(i)}{max\{a(i), b(i)\}}, \tag{7}$$

where $a(i)$ is the average $Ed$ between the $i$[th] member in cluster $A$ and the other members of this cluster; $b(i)$ is the average $Ed$ between member $i$ and the members in its second closest cluster $B$.

The silhouette ranges between $-1$ and 1. If $s(i)$ close to 1, the member $i$ is well-assigned to cluster $A$. If $s(i)$ is close to 0, the object can be in-between of $A$ and $B$. If $s(i)$ is close to $-1$, the object is badly assigned to its cluster and on average closer to members of $B$ [70].

### 4.3. Automatic Generation of Time Series of Water Levels

Based on the results of the unsupervised classification, time series of water levels are generated from RA data over the classes corresponding to open water and inundated areas. A time series is generated if there are at least five non-flagged altimetry heights within a distance of 3 km non-separated between each other by more than 1 km. The minimum distance between two different VSs is 3 km.

### 4.4. Validation

Two types of evaluation of the classification of altimetry data are performed. Global validation is performed against the land cover map of the Cuvette Centrale from [40]

considered as the reference. The confusion matrix is estimated when comparing the classes derived from the k-means clustering technique to the classes of land types.

Local validation parameters (bias, RMSE, R) were also performed comparing the time series of water level automatically produced using the results from the unsupervised to time series of water levels from Hydroweb.

### 4.5. Surface Water Volume Estimates

Maps of surface water levels were obtained combining inundation extent derived from the land cover map of the Cuvette Centrale [40] with radar altimetry-based water levels from ENVISAT and Jason-2. The methodology defined in [22,23] and applied to different types of satellite images, such as SAR [22], multispectral [71] and passive microwaves [72], was used in this study. It consists in interpolating altimetry-based water stages over the inundated areas using an inverse distance weighting (IDW) technique. The wetlands inundation map was used to simulate the inundation extent over 2006–2010 as the PALSAR images used to achieve this inundation map do not allow to reach a monthly temporal resolution. During a low water period (February to April and July–August), water extent is limited to open water and permanent inundation classes (numbered 1 and 2 in Section 3.2). For the first high water period and the rising and decline of the second and larger high water period (May–June and September–October), seasonally flooded areas during the two flood pulses (class 3) are also considered, and for the peak of high water period (November–January), all classes related to water (i.e., the previously mentioned and the seasonally flooded areas during a short time corresponding to the maximum of the largest flood pulse, class 4) are merged. A map of minimum water levels recorded during the whole common observation period between SAR and altimetry was computed using a hypsometric approach to take into account the difference of altitude between the river and the floodplain (see [23] for more details on the method).

## 5. Results

### 5.1. Automatic Generation of Time Series of Water Levels

The optimal number of classes of backscattering coefficients was chosen by analyzing the Calinski–Harabasz criterion and the silhouette index from the number of classes ranging from 2 to 10 for both ENVISAT and Jason-2 data. The evolution of these indices as a function of the number of class is presented in Figure 2 for ENVISAT and Jason-2 backscattering. As silhouette coefficients are far from 1 for both ENVISAT (maximum value of 0.58) and Jason-2 (maximum of 0.51) backscattering coefficients, they cannot be used for determining the optimal number of classes from altimetry backscattering coefficients. On the contrary, $I_{CH}$ presents a clear peak for both ENVISAT and Jason-2 for 4 and 3 classes respectively (Figure 2). Nevertheless, very similar values of $I_{CH}$ (varying from less than 5% from the maximum value) can be observed for a number of classes ranging from 3 to 5 for both altimeters. In the following, all the results are be presented using the number of classes varying from 3 to 5.

### 5.2. Unsupervised Classification Results

K-means unsupervised classification was applied to ENVISAT and Jason-2 backscattering at Ku-band considering 3 to 5 classes. Figures 3 and 4 present the seasonal distribution of backscattering at Ku-band from ENVISAT and Jason-2 respectively for class numbers ranging from 3 to 5. The different classes obtained from the k-means approach exhibit a quite similar temporal behavior with a maximum occurring in December, a minimum in August, a secondary maximum in May and a secondary minimum in March. This seasonal change is in phase with the temporal variations in the water levels and discharge of the Congo River (e.g., [38,65]) but in opposite phase with the backscattering at L-band from PALSAR-1 images [38]. Larger seasonal amplitudes are observed for the classes characterized by a higher backscattering. Better separability is observed for 3 and 4 than for 5 classes (i.e., less overlap of the *std*). In this latter case, the distances between the center

of the consecutive classes can be lower than the sum of the standard deviations of these two classes. This lower separability affects more the classes with the lower backscattering coefficients and more the Jason-2 data than the ENVISAT data.

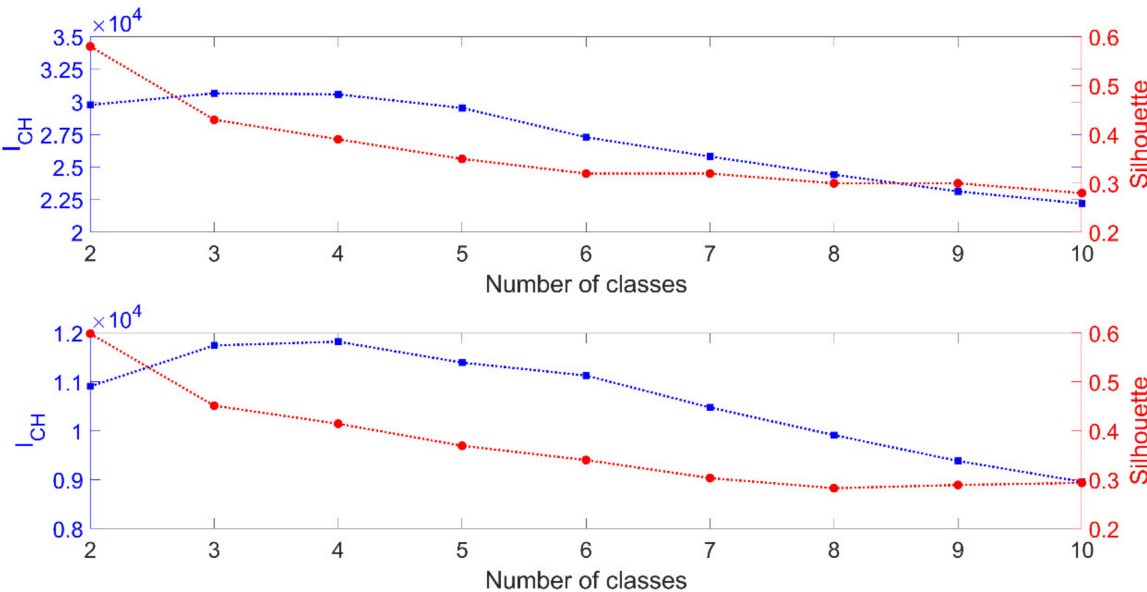

**Figure 2.** Evolution of the Calinski–Harabasz criterion ($I_{CH}$ in blue) and the silhouette (in red) criterion as a function of the number of classes for ENVISAT (upper panel); Jason-2 (lower panel) backscattering coefficients.

The following classes are present in the classification of the Cuvette Centrale used in this study:

(i)　Open water;
(ii)　Permanently flooded forests;
(iii)　Seasonally flooded forests during the two Congo River flood pulses and located alongside the river;
(iv)　Seasonally flooded forests during a short time corresponding to the maximum of the largest flood pulse, farther from the river;
(v)　Non-flooded forests.

From the results obtained in Section 5.1, confusion matrices were obtained using 3 to 5 classes from the radar altimetry backscattering datasets. They are presented in Table A1 (3 classes), Table A2 (4 classes) and Table 1 (5 classes) for ENVISAT and Table A3 (3 classes), Table A4 (4 classes) and Table 2 (5 classes) for Jason-2.

**Table 1.** Confusion matrix between 5 classes of RA backscattering coefficients from ENVISAT and the 5 land type classes identified on the study sites.

| ENVISAT Class | Open Water | Flood. Perm. | Flood. Seas. Long | Flood. Seas. Short | Non-Flood. |
|---|---|---|---|---|---|
| 1 | 0.42 | 0.29 | 0.21 | 0.07 | 0.01 |
| 2 | 0.19 | 0.26 | 0.36 | 0.17 | 0.02 |
| 3 | 0.05 | 0.09 | 0.43 | 0.34 | 0.09 |
| 4 | 0.00 | 0.03 | 0.34 | 0.49 | 0.14 |
| 5 | 0.01 | 0.01 | 0.15 | 0.57 | 0.26 |

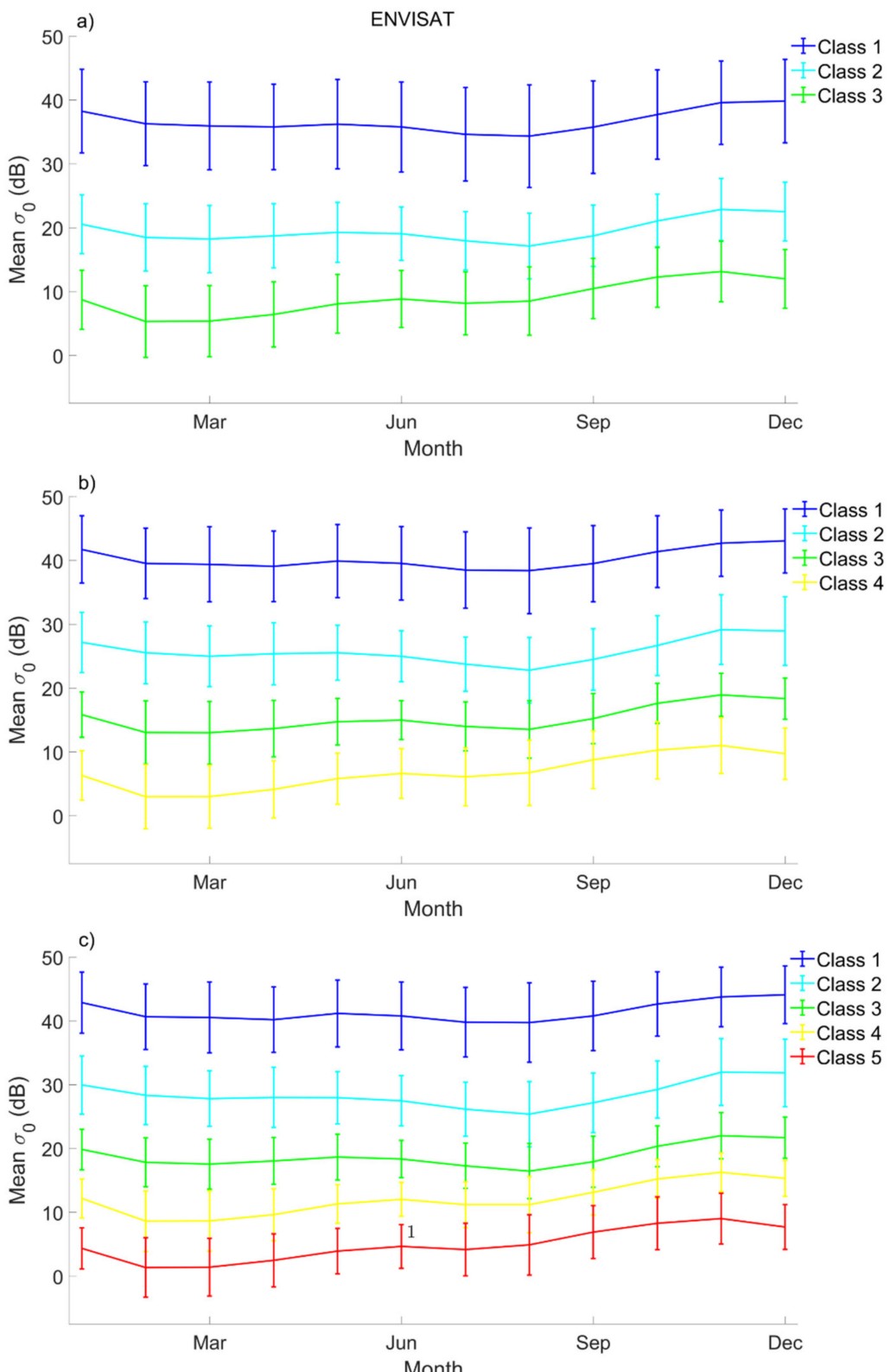

**Figure 3.** Seasonal variations in the mean backscattering coefficient and its associated deviation at Ku-band from ENVISAT (2003–2010) for each class when considering (**a**) 3, (**b**) 4, (**c**) 5 classes.

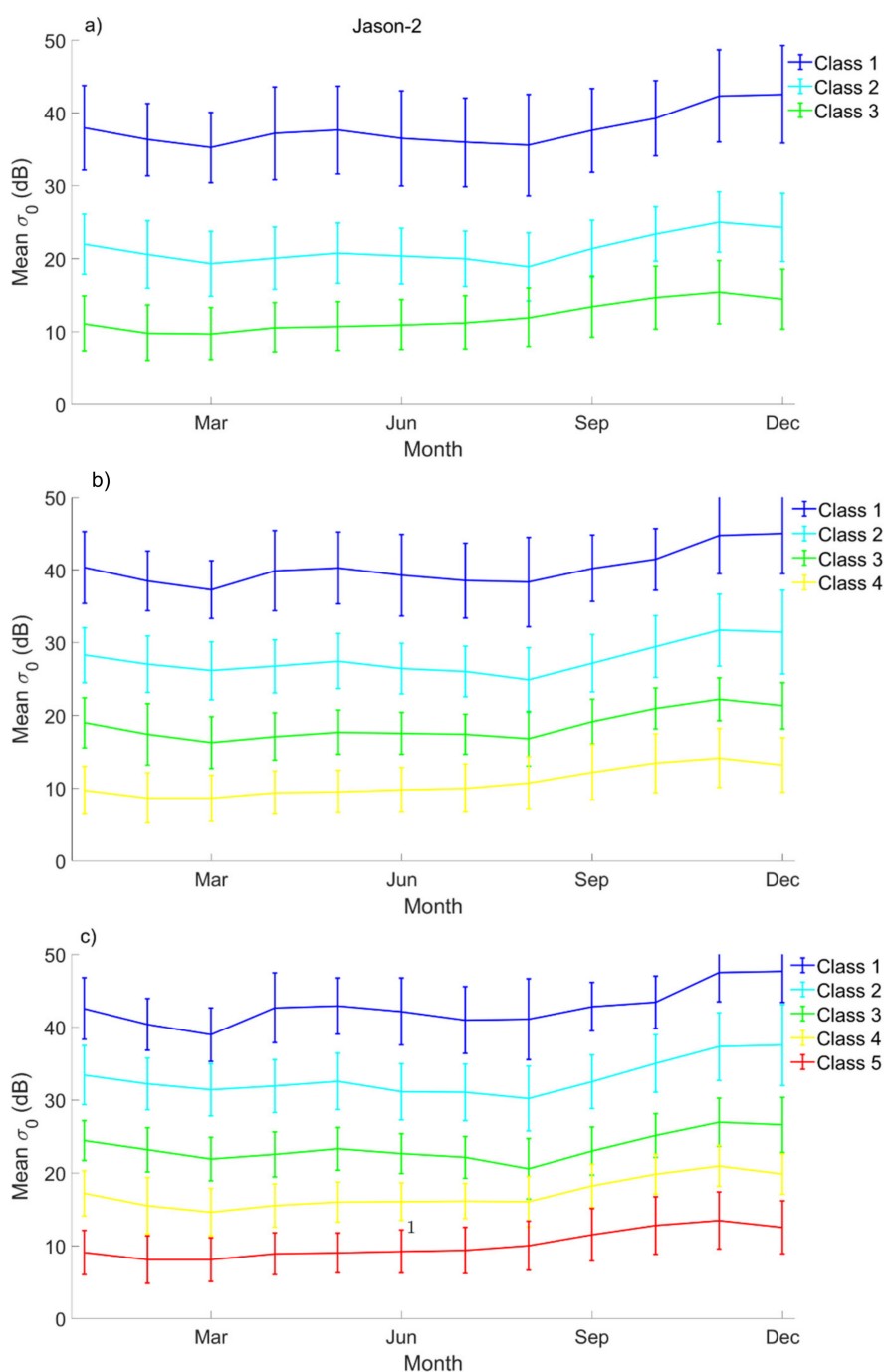

**Figure 4.** Seasonal variations in the mean backscattering coefficient and its associated deviation at Ku-band from Jason-2 (2008–2016) for each class when considering (**a**) 3, (**b**) 4, (**c**) 5 classes.

**Table 2.** Confusion matrix between 5 classes of RA backscattering coefficients from Jason-2 and the 5 land type classes identified on the study sites.

| Jason-2 Class | Open Water | Flood. Perm. | Flood. Seas. Long | Flood. Seas. Short | Non-Flood. |
|---|---|---|---|---|---|
| 1 | 0.60 | 0.21 | 0.17 | 0.01 | 0.01 |
| 2 | 0.21 | 0.29 | 0.31 | 0.17 | 0.02 |
| 3 | 0.01 | 0.10 | 0.47 | 0.33 | 0.09 |
| 4 | 0.00 | 0.01 | 0.38 | 0.45 | 0.16 |
| 5 | 0.00 | 0.00 | 0.10 | 0.64 | 0.26 |

Radar altimetry backscattering classes do not have the same distribution as land type classes. Maximum values range between 0.6 and 0.65, e.g., Jason-2 class 1 data are distributed at 0.6 and 0.62 in the open water class when considering 4 and 5 classes, respectively, at 0.62 and 0.64 for short duration flooded forests (Table 2 and Table A4 respectively). These results can appear low compared with those obtained for images classification. However, as the footprint of the radar altimeter has a radius of several kilometers, it encompasses very heterogenous surfaces in terms of roughness, soil and vegetation types, moisture content and presence or not of surface water.

Previous studies analyzing radar altimetry backscattering coefficients over land surfaces showed that high values of backscattering are observed over rivers and floodplains and low values over vegetation [32,35,36,64,73].

When merging results from predominantly water and vegetation, we obtain:

- The results 0.89, 0.91 and 0.92 for ENVISAT 0.89, 0.93, 0.98 for Jason-2 when considering the sum of the backscattering values in class 1 over open water, permanently and long duration flooded forests;
- The results 0.72, 0.89, 0.93 for ENVISAT and 0.85, 0.79, 0.90 for Jason-2 when considering the sum of the backscattering values in the last class over short duration flooded and non-flooded forests.

The spatial distribution of the backscattering classes is presented for ENVISAT and Jason-2 over the whole Cuvette Centrale in Figure 5 and a smaller area in Figure 6 to illustrate the good agreement between the unsupervised classification and the land types.

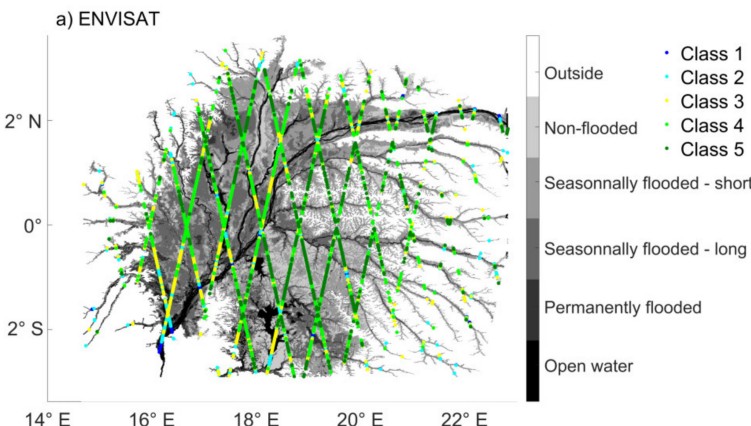

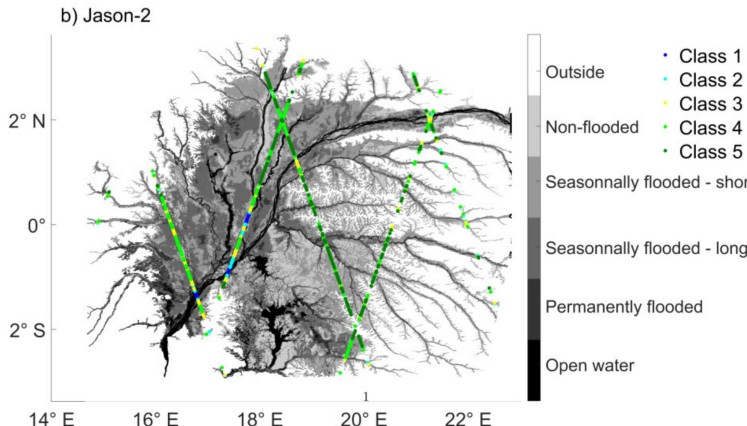

**Figure 5.** Spatial distribution of the backscattering classes (Ku band) from (**a**) ENVISAT and (**b**) Jason-2 in the Cuvette Centrale of Congo.

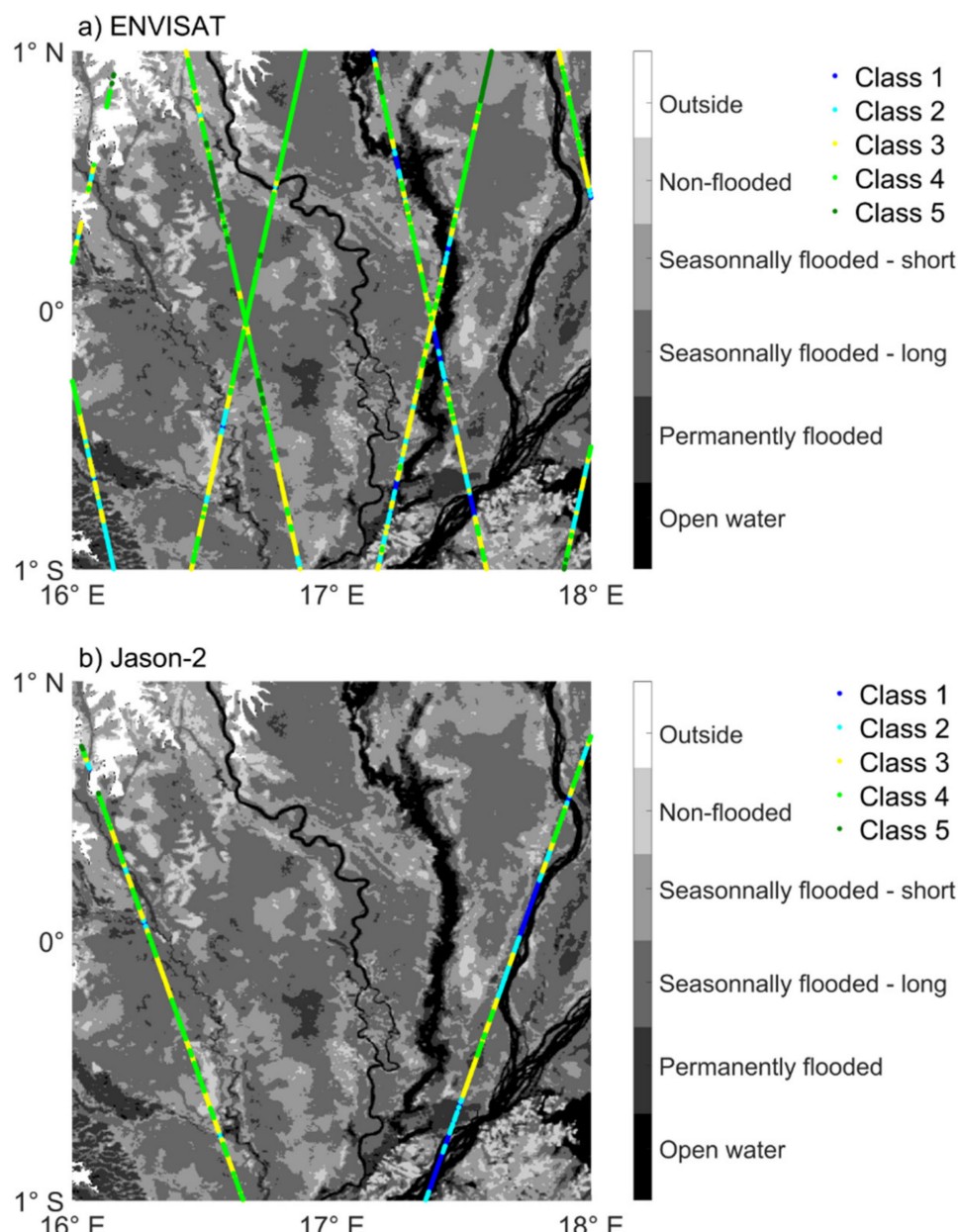

**Figure 6.** Same as Figure 5 for a smaller area.

### 5.3. Time Series of Water Levels and Volumes

A total of 250 and 358 time series of water levels were automatically generated over the Cuvette Centrale of Congo using Jason-2 and ENVISAT data respectively for five classes of backscattering (Figure 7). Comparisons were performed between them and the closest location (below 5 km) of a Hydroweb VS. Bias, RMSE and R were estimated for 24 Jason-2 and 32 ENVISAT VSs (Figure 8). Very good agreement was found between the automatically generated time series and the Hydroweb ones with R generally higher than 0.95 (18 out of 24 for Jason-2 and 31 out of 32 for ENVISAT), RMSE lower than 0.25 m (15 out of 24 for Jason-2 VS, 22 out 32 for ENVISAT VSs) and bias lower than 0.25 m (22 out of 24 for Jason-2 VS, 26 out of 32 for ENVISAT VSs).

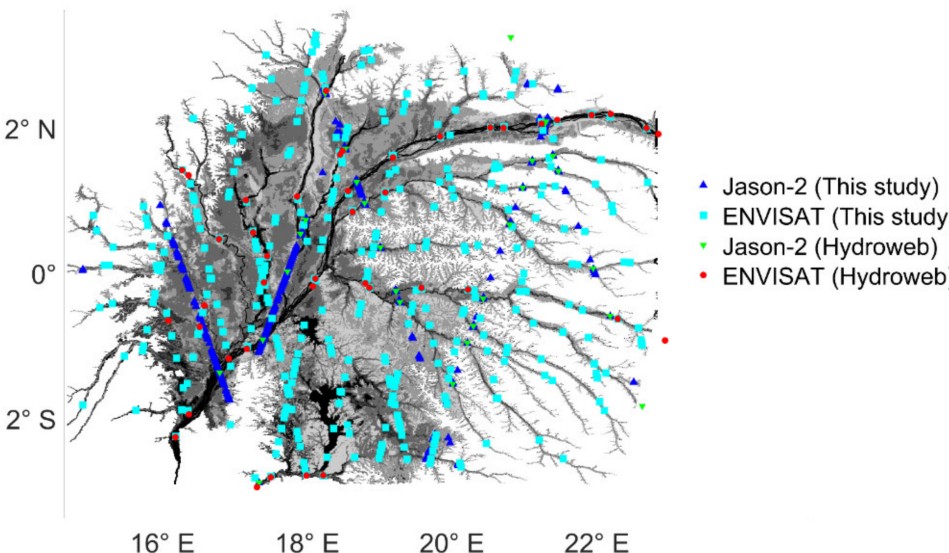

**Figure 7.** Locations of the VSs automatically created on the Jason-2 (blue triangles) and ENVISAT (light blue squares) ground-tracks and of the VSs from Hydroweb (Jason-2, green triangles; and ENVISAT, red circles) used for comparison.

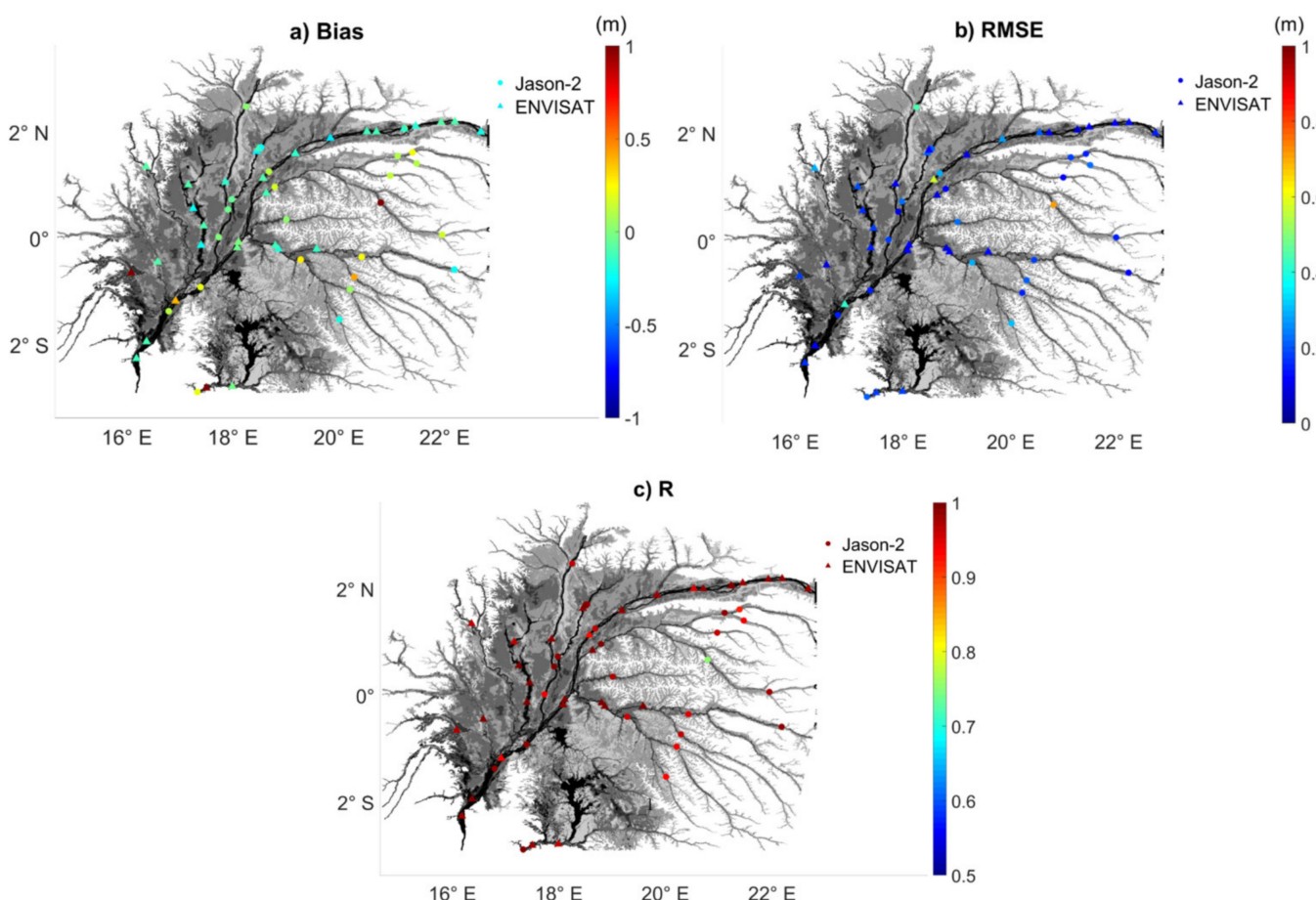

**Figure 8.** Comparison between altimetry-based of water levels from Hydroweb and automatically generated (this study) in terms of (**a**) bias, (**b**) RMSE and (**c**) r. VS locations are represented using either a dot (Jason-2) or a triangle (ENVISAT).

### 5.4. Surface Water Storage

Monthly maps of surface water levels and associated water volume variations were derived interpolating altimetry-based water levels on the wetland extent at a monthly temporal resolution. To avoid differences caused by intermission biases, these maps were obtained only using ENVISAT VSs. The VSs used were the ones automatically derived over either the rivers or the rivers and the floodplains. The associated surface water volume variations were computed in both cases. The annual amplitude for the largest flood period (November–December) is around four times higher (above 800 km$^3$) and the flood peak occurs 1 to 2 months later (December–January) only using VSs on rivers then using VSs on both the rivers and the floodplains (Figure 9). These differences of volume are caused by differences in spatial patterns of the water stage. Maps of the minimum height were obtained from ENVISAT. The minimum height is either the minimum of the water level during the observation period over rivers and permanently flooded pixels, or the bottom topography of floodplains over non-permanently inundated areas. In Figure 10 are presented maps based on the VSs automatically created over the rivers and the floodplains (a), or based on VSs manually created over the rivers and made available by Hydroweb (b). As it can be seen from their difference (c), the minimum height is quite similar on the center of the Cuvette Centrale, and increases up to ~30 m on the edges of the study area. Considering the mean annual amplitude, either automatically created over rivers and floodplains (d) or manually created over rivers only (e), differences up to ~10 m, positive in the north and negative in the south can be observed (f).

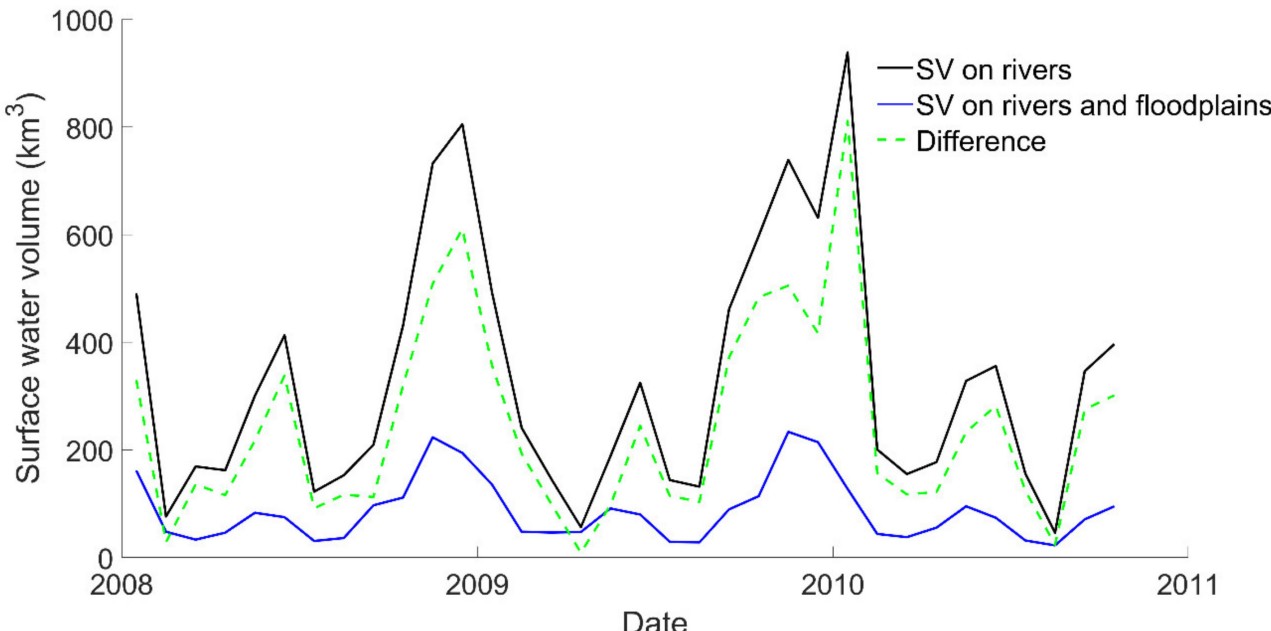

**Figure 9.** Time series of surface water storage of the Cuvette Centrale of Congo obtained combining the surface water extent and the time series of water levels from the ENVISAT VSs on the rivers (black), on the rivers and floodplains (blue). Their difference is presented in dashed green.

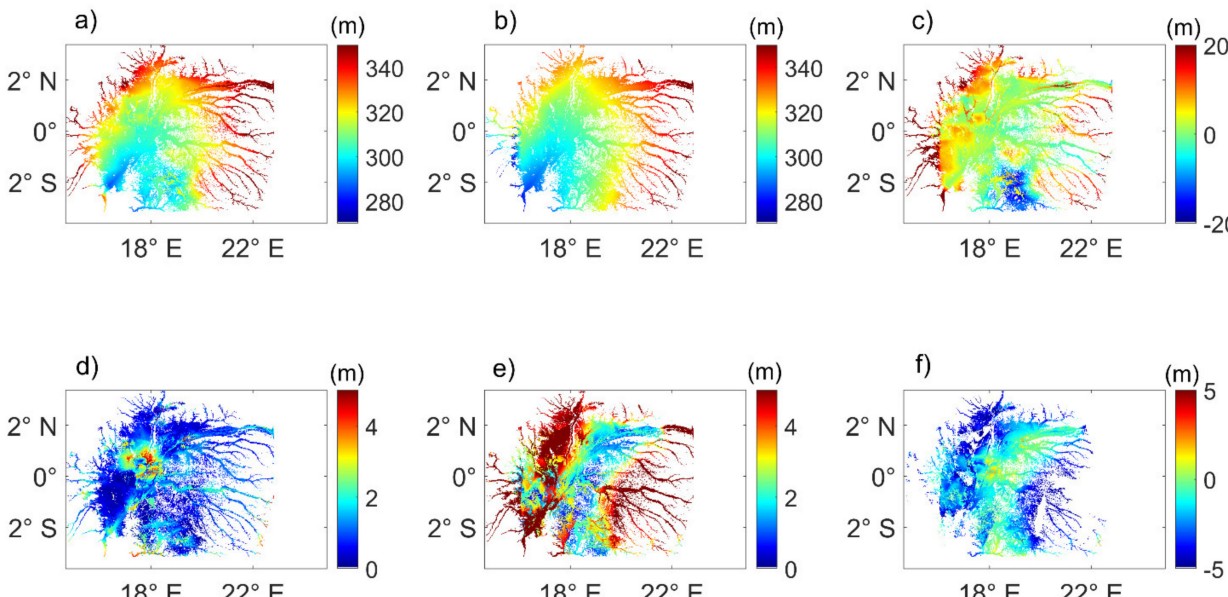

**Figure 10.** Maps of minimum of height from ENVISAT based on the VSs (**a**) automatically created over the rivers and the floodplains or (**b**) manually created over the rivers and made available by Hydroweb, and (**c**) their difference. Maps of mean annual amplitude using the VSs (**d**) over the rivers and floodplains, (**e**) only over the rivers, and (**f**) their difference.

## 6. Discussion

### 6.1. Identification of Open Water on RA Ground-Tracks

The results of the classification of the RA backscattering coefficients show that this parameter can be used to efficiently discriminate between water and vegetation under both ENVISAT and Jason-2 ground-tracks. Contrary to SAR or multi-spectral images, this type of sensor seems to be unable to discriminate between open water and water under vegetations in the floodplains (see the results of the confusion matrices presented in Tables 1, 2 and A1–A4). This is most likely due to:

(i)   The size of the illuminated area at Ku-band (several km of diameter in low resolution mode (LRM)), the scene present in the altimeter footprint is very heterogenous and encompasses both rivers and the surrounding floodplains;

(ii)  The RA sensor is acquiring data at nadir. As a consequence, the power backscattered by the water dominates the radar echo. RA backscattering are much stronger over rivers and floodplains than over any other types of land cover (e.g., [32,36,37]). Even under forest canopy, water levels can be retrieved (e.g., [22] and this study).

All the classes exhibit a similar temporal behavior even for the classes corresponding to non-inundated areas. The smaller seasonal cycle observed for these classes can be attributed to an increase in soil moisture during the wet periods, which a change in the dielectrical properties of the soil, and hence a rise in $\sigma_0$ which can be detected even under a dense vegetation cover [35–37,74].

This good ability allowed to automatically generate time series of water levels on the RA ground-tracks; 358/250 VSs were generated on the ENVISAT/Jason-2 ground-tracks, respectively. The ones on the river were validated against manually created time series of water levels from Hydroweb due to the lack of in-situ data for validation. Very similar results were obtained when using the approach presented in this study versus the manual creation of VSs (RMSE $\leq$ 0.25 m in more than 80%/90% of the 32/24 cases and R $\geq$ 0.95 in more than 95%/75% of the same cases for ENVISAT and Jason-2, respectively). The time series of the anomaly of water levels from ENVISAT and Jason-2, averaged over all the stations located on the river and the permanently inundated areas, and over non-permanently inundated areas are presented on Figure 11. The temporal variations are consistent with the hydrological regimes of the area: a first maximum in April–May, a secondary larger one

in November–December, as well as the annual amplitude around 3–4 m (e.g., [65,67,68]) and lower over the floodplains [47], as it can be seen when considering the average $\pm$ 1 *std*. The lower mean annual amplitude is observed on time series from ENVISAT (Figure 11a,c) than Jason-2 (Figure 11b,d). This is most likely due to the longer temporal revisit period of ENVISAT (35 days) than Jason-2 (10 days) and the higher number of VSs on the ENVISAT ground-tracks (358) than Jason-2 (250).

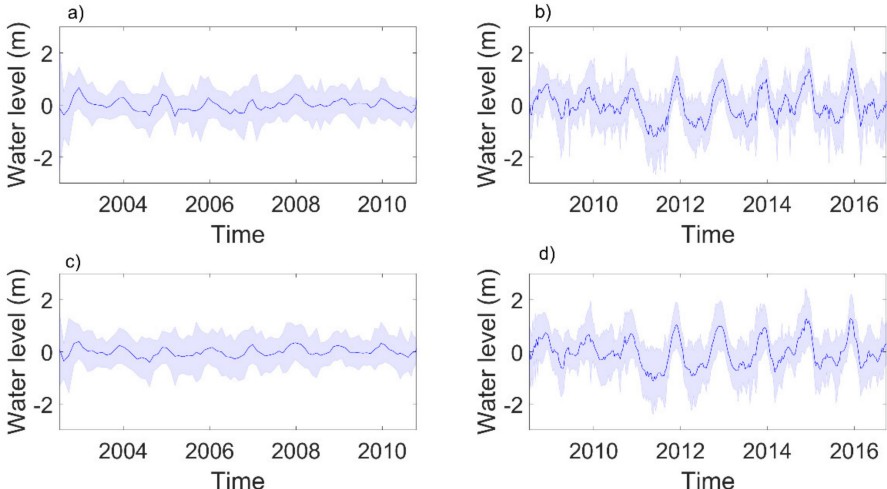

**Figure 11.** Time series of anomaly of water levels averaged over all the stations located on the river and the permanently inundated areas from (**a**) ENVISAT and (**b**) Jason-2, and over the non-permanently inundated areas from (**c**) ENVISAT and (**d**) Jason-2.

### 6.2. Impact on Anomaly of Surface Water Storage

The densification of the network of VSs with the definition of new VSs over the floodplains and wetlands offer a unique opportunity to better understand the dynamics of this hydrological reservoir. Two parameters are used to characterize the surface reservoir: the discharge and the storage. The combination of surface extent with water level variations provides access to the surface water storage changes (e.g., [22,71]). In most of the river basins, time series of water level from RA are only recorded at VSs over rivers. This study offers an opportunity to measure the impact of the inclusion of VSs over floodplains on the estimates of surface water storage variations. Even if the real time series of inundation extent is not considered here, realistic monthly variations in this parameter were used. Note that a recent study showed that the use of a climatology of surface water extent was sufficient to retrieve accurate variations in surface water storage [75]. Our study showed that the differences in bottom topography of the floodplains (Figure 10a–c) and in the patterns of the annual amplitude of water level (Figure 10d–f) have large consequences on the amplitude of the variations in surface water storage in the Cuvette Centrale of Congo. The lack of VSs on rivers for the upstream parts in the northwest of the study area leads to an underestimate of the bottom topography from 10 to 30 m, and an overestimate of the annual amplitude above 5 m, causing an overestimate of the storage by a factor of four. Besides, spatial patterns of the annual amplitude are more consistent with the spatial distribution of the wetlands in the Cuvette Centrale. For instance, if we consider the region centered on 18° E and from 0 to 2° N, high annual amplitude is observed in the Ngiri and Eulonga basins (below 1° N), a close to zero annual amplitude above, in the Upper Ubangi basin when using the VSs on the floodplains in the interpolation (Figure 10d), in good agreement with the map of the probability of wetlands [46]. On the contrary, high annual amplitude is found on the whole area when only using the VSs on the rivers (Figure 10e). These differences account for the change in amplitude and temporality (Figure 9) between the study cases (VSs only over the rivers or on the rivers and the floodplains). The influence of the larger changes in water levels observed over the rivers is limited when increasing the number of VSs in the IDW both on the storage and its seasonality (e.g., the signal is

dominated by the largest floodplains and the time-lag of the seasonal cycle of the Congo River tributaries).

## 7. Conclusions

Radar altimetry is a unique tool for measuring water levels over inland waterbodies, and especially over the non-gauged wetlands and floodplains. Identifying the presence of water in wetlands and floodplains (where and when) makes difficult the construction of VSs over these environments. In this study, applying a k-means unsupervised classification approach to RA backscattering coefficients allows to accurately identify the flooded (good detection over 90%) and non-flooded areas over a complex equatorial area, the Cuvette Centrale of Congo, where floodplains are mostly covered with vegetation. Contrary to other types of satellite products, RA, due to the size of its footprint (several kilometers of diameter at Ku-band), seems unable to discriminate between open water and permanently flooded areas or to provide information on the type of vegetation covering the non-flooded area. Using the classification results, time series of water levels were automatically created over the rivers and the floodplains of the Cuvette Centrale of Congo (358/250 on the ENVISAT and Jason-2 ground-tracks, respectively). The resulting time series exhibit temporal variations in good agreement with the hydrological regime of the study area in terms of temporal variations and annual amplitude. Very similar results to the manually created time series of water levels were obtained: (RMSE $\leq$ 0.25 m in more than 80%/90% of the 32/24 cases and R $\geq$ 0.95 in more than 95%/75% of the same cases for ENVISAT and Jason-2, respectively). The densification of the VSs network by including the floodplains has a strong implication on the water volume estimate.

Owing to the larger number of VSs used when considering the VSs on the floodplains, the interpolation of the water level maps is more constrained, and the spatial patterns of annual amplitudes are more consistent with the information on the wetland extent. With the availability of data from recent RA missions operating in SAR mode such as SENTINEL-3A and B, better discrimination based on the backscattering can be expected, allowing to continue the monitoring with a better accuracy of the time series of water levels over the floodplains. This approach is likely to be applied to any other extensive wetlands or floodplains and will contribute to a better estimate, in combination with satellite images, to the changes in surface water storage over land. As the high precision altimetry era started in the early/mid 1990s with the launch of Topex/Poseidon (1992) and ERS-1/2 (1991 and 1995, respectively), a long-term record of almost 30 years of surface water storage variations can soon be available. All this information will have a large importance for better understanding the hydrological processes in the floodplains and wetlands and for the comparison with data acquired by the future NASA/CNES SWOT mission, to be launched in 2022, which will the first to estimate water levels in a swath.

**Author Contributions:** Conceptualization, F.F., J.B. and V.G.; methodology, F.F., J.D., L.B., F.S.; processing, F.F., P.Z. and R.B.; data curation, F.B., N.B., F.S.; writing—original draft preparation, F.F.; writing—review and editing, all authors; funding acquisition, F.F., J.B., V.G. and J.D. All authors have read and agreed to the published version of the manuscript.

**Funding:** This research was funded by CNES TOSCA grants number CASCHMIR and SWHYM. The APC was offered by MDPI.

**Institutional Review Board Statement:** Not applicable.

**Informed Consent Statement:** Not applicable.

**Data Availability Statement:** Radar altimetry data can be found here: http://ctoh.legos.obs-mip.fr/applications/land_surfaces/altimetric_data/altis/altis. Time series of water levels can be found here: http://hydroweb.theia-land.fr/. Accessed on 1 April 2019.

**Acknowledgments:** We thank Sylvie Gourlet-Fleury (CIRAD) for helpful discussion during the preparation of the manuscript. We also thank three anonymous reviewers for their meaningful comments which helped us improving the quality of our manuscript.

**Conflicts of Interest:** The authors declare no conflict of interest.

## Appendix A

**Table A1.** Confusion matrix between 3 classes of RA backscattering coefficients from ENVISAT and the 5 land type classes identified on the study sites.

| ENVISAT Class | Open Water | Flood. Perm. | Flood. Seas. Long | Flood. Seas. Short | Non-Flood. |
|---|---|---|---|---|---|
| 1 | 0.33 | 0.30 | 0.26 | 0.10 | 0.01 |
| 2 | 0.06 | 0.10 | 0.41 | 0.34 | 0.09 |
| 3 | 0.01 | 0.02 | 0.25 | 0.53 | 0.19 |

**Table A2.** Confusion matrix between 4 classes of RA backscattering coefficients from ENVISAT and the 5 land type classes identified on the study sites.

| ENVISAT Class | Open Water | Flood. Perm. | Flood. Seas. Long | Flood. Seas. Short | Non-Flood. |
|---|---|---|---|---|---|
| 1 | 0.40 | 0.29 | 0.22 | 0.08 | 0.01 |
| 2 | 0.08 | 0.17 | 0.43 | 0.26 | 0.06 |
| 3 | 0.00 | 0.04 | 0.42 | 0.41 | 0.13 |
| 4 | 0.00 | 0.00 | 0.11 | 0.64 | 0.25 |

**Table A3.** Confusion matrix between 3 classes of RA backscattering coefficients from Jason-2 and the 5 land type classes identified on the study sites.

| Jason-2 Class | Open Water | Flood. Perm. | Flood. Seas. Long | Flood. Seas. Short | Non-Flood. |
|---|---|---|---|---|---|
| 1 | 0.41 | 0.25 | 0.24 | 0.09 | 0.01 |
| 2 | 0.01 | 0.07 | 0.47 | 0.35 | 0.10 |
| 3 | 0.00 | 0.00 | 0.15 | 0.62 | 0.23 |

**Table A4.** Confusion matrix between 4 classes of RA backscattering coefficients from Jason-2 and the 5 land type classes identified on the study sites.

| Jason-2 Class | Open Water | Flood. Perm. | Flood. Seas. Long | Flood. Seas. Short | Non-Flood. |
|---|---|---|---|---|---|
| 1 | 0.48 | 0.26 | 0.19 | 0.06 | 0.01 |
| 2 | 0.15 | 0.21 | 0.39 | 0.21 | 0.04 |
| 3 | 0.01 | 0.05 | 0.40 | 0.42 | 0.12 |
| 4 | 0.01 | 0.02 | 0.18 | 0.57 | 0.22 |

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
