# Peer review of "Automatic Detection of Inland Water Bodies along Altimetry Tracks for Estimating Surface Water Storage Variations in the Congo Basin"

_remotesensing, doi:10.3390/rs13193804_

Round 1
Reviewer 1 Report
This paper presents solid research on automatic detection of inland water bodies using altimetry. I recommend the paper to be accepted after addressing revisions below.
General: It’s very hard to find some Figures when I read the manuscript. Please resize and place these figures with the text, so it will be easy for readers to follow.
Page 1 Line 45: ‘import’ should ‘important’
Page 2 Line 53: InSAR is usually for ‘Interferometric Synthetic Aperture Radar’
Page 3 Line 130: Some places looks like two ‘space’, please double check
Page 5 Line 183: delete unnecessary space
Page 5 Line 207: ‘where’ may be replaced by ‘and’
Page 6 Line 215-216: Delete one ‘respectively’
Page 6 Line 222: Add space for a new paragraph
Page 6 Line 240: ‘sigma0’ missed
Page 6 Line 244: Please use the correct equation for (6)
Page 7 Line 260: Add space for a new paragraph
Page 7 Line 268: ‘virtual stations (VS)’ was shown before
Page 7 Line 286: Please correct ‘July August’
Page 8 Line 312: Please use subscript for ‘CH’
Page 9 Table 1: The summation of 5 land types for class 1 is not 1
Page 12 Line 378: Extra space before ‘Bias’
Page 12 last paragraph: please use same format for 10a – 10f. no need to repeat ‘Figure’.
Page 17 Line 441: Delete space before ‘one std’
Page 17 Line 449: ‘RMSE ≤ 0.25 m’ is repeated; Delete one of them or add a ‘/’
Page 18 Line 490: ‘makes’?
Page 18 Line 492: ‘allows’?
Page 19 Line 503: ‘RMSE ≤ 0.25 m’ is repeated; Delete one of them or add a ‘/’
Page 20 Line 541: should be ‘4 classes’
Author Response
Pease find our reponses to your comments in the enclosed file.

Reviewer 2 Report
Comments on Automatic detection of inland water bodies along altimetry tracks for estimating surface water storage variations in the Congo Basin:
- In the last paragraph of the Introduction, the authors should clearly mention the weakness point of former works (identification of the gaps) and describe the novelties of the current investigation to justify us the paper deserves to be published in this journal.
- In the Abstract, cite these recent useful papers on global warming and climate change which impact on surface water storage variations to improve the literature and to show the importance of your work:
Projected Change in Temperature and Precipitation Over Africa from CMIP6
Global surface temperature: A new insight
- Discuss more the comparison between altimetry‐based of water levels from Hydroweb and automatically generated.
- Focus on the main reasons for the variations of the Time series of surface water storage of the Congo Cuvette Centrale obtained combining the surface water extent and the time‐series of water levels from the ENVISAT SV.
- How can expand the results in other regions with similar/different climates?
- At the end of the manuscript, explain the implications and future works considering the outputs of the current study.
- The quality of the language needs to improve by a native English speaker for grammatically style and word use.
Author Response

(The authors gave the same response as above.)

Reviewer 3 Report
- Line 103: Study area. Please briefly describe the land use/ land cover in the study area. It is helpful for reader to understand your study area better.
- Line 308: Very similar values of ICH (varying from less than 5% from the maximum value) can be observed for a number of classes ranging from 3 to 5 for both altimeters. So, this study is using number of classes varying from 3 to 5. Why to choose less than 5%? How to decide? How about 10%?
- Line 318: The different classes exhibit a quite similar temporal behavior with a maximum occurring in December, a minimum in August. However, many classes in Figure 3 and 4 show that the minimum was in March not in August. Is it correct? Please explain this point.
- Line 392: The annual amplitude for the largest flood period (November-December) is around four times higher and the flood peak occurs 1 to 2 months later (December-January) only using SV on rivers than using SV on both the rivers and the floodplains (Figure 9). Why the flood peak occurs later so long? Please explain it more detail.
- Line 480: Where is the Figure 9d? Is it wrong? Figure 10d?
- Line 482: It is the same mistake. Where is the Figure 9e?
Author Response

(The authors gave the same response as above.)

Round 2
Reviewer 1 Report
The authors addressed the main concerns from the reviews, and the revised version looks good. I recommend it to be published as it is.
Reviewer 2 Report
I have no further comment.
Reviewer 3 Report
Accept in present revised form.